# Factors behind Antibiotic Therapy: A Survey of Primary Care Pediatricians in Lombardy

**DOI:** 10.3390/ijerph21081091

**Published:** 2024-08-18

**Authors:** Pier Mario Perrone, Marina Picca, Romeo Carrozzo, Carlo Virginio Agostoni, Paola Marchisio, Gregorio Paolo Milani, Silvana Castaldi

**Affiliations:** 1Department of Pathophysiology and Transplantation, University of Milan, 20122 Milan, Italy; paola.marchisio@unimi.it; 2Department of Clinical Sciences and Community Health, University of Milan, 20122 Milan, Italy; carlo.agostoni@unimi.it (C.V.A.); gregorio.milani@unimi.it (G.P.M.); 3Italian Primary Care Paediatrics Society (SICuPP), 20126 Milan, Italy; 4Pediatric Emergency Units, Fondazione Ca’ Granda Policlinico di Milano, 20122 Milan, Italy; 5Pediatric Infectious Disease Units, Fondazione Ca’ Granda Policlinico di Milano, 20122 Milan, Italy; 6Department of Biomedical Sciences for Health, University of Milan, 20133 Milan, Italy; silvana.castaldi@unimi.it; 7Quality Units, Fondazione Ca’ Granda Policlinico di Milano, 20122 Milan, Italy

**Keywords:** antimicrobial resistance, antibiotic prescription, primary care pediatricians

## Abstract

Background: Antimicrobial resistance represents one of the most significant future health challenges in terms of both clinical and economic impacts on healthcare systems. The reason behind this issue is the misuse of antibiotics for the treatment of non-bacterial pathologies. The objective of this study is to investigate the factors underlying antibiotic prescription in pediatricians in the Lombardy region. Methods: The study was conducted by means of a 32-item questionnaire that investigated both pediatricians’ knowledge of antimicrobial resistance and the factors determining the choice to prescribe antibiotic therapy. Results: A total of 253 pediatricians participated in the survey. Most participants (71.6%) reported as highly relevant the need for a national plan against AMR. However, approximately half of the respondents declared the phenomenon of AMR as uncommon in pediatric settings. Among the identified associated factors, diagnostic uncertainty was associated with a stronger fear of legal repercussions and the influence of parental pressure when prescribing antibiotics. Conclusions: The inability to diagnose the bacterial origin of an infection might be the primary driver of prescribing choices, rather than other non-clinical factors, such as parental demands or a fear of lawsuits.

## 1. Introduction

Antimicrobial resistance (AMR) is one of the main issues in public health. Defined as the resistance of microorganisms to an antimicrobial agent to which they were initially sensitive, AMR resulted in 1.27 million global deaths in 2019 and contributed to 4.95 million deaths [1]. Furthermore, AMR is responsible for financial losses due to the use of more expensive antibiotics, specialized equipment, longer hospital stays and isolation procedures for patients [2,3]. In Europe, AMR accounts for an economical burden of EUR 1.5 billion, with more than EUR 900 million due to direct hospitalization costs [4].

For these reasons, several organizations have expressed the need for antimicrobial stewardship programs in order to address this problem [5,6,7,8,9,10,11]. Due to the various healthcare systems set up, different countries have implemented multiple programs characterized by the strict control of antimicrobial use and infection prevention in hospital environments [12]. However, this approach does not take into account the impact of general practitioner (GP) activity on antibiotic prescription. In Italy, GP and primary care pediatricians account for almost 90% of antibiotic prescriptions [13]. Moreover, inappropriate prescriptions amount to about 25% in this setting. This represents one of the most significant—if not the primary—factors contributing to this phenomenon, resulting in notable implications for both community health and healthcare organizations at the hospital level.

Despite the importance of this topic and the above-mentioned data, there are few studies in the literature aimed at understanding the reasons that a primary care pediatrician might prescribe antibiotics inappropriately [13,14].

Notably, a systematic review performed by Sijbom in 2023 tried to assess the influence of several factors on inappropriate prescription [15]. The results showed a strict connection between patients and clinician-related determinants without a specific element that could lead to erroneous or inappropriate prescription. However, despite this great interest in the subject of factors influencing the willingness to prescribe antibiotics, there are no examples in the literature of attempts to analyze their simultaneous influence on prescription, instead assessing the main ones or the different weights of some of them.

The primary aim of this study is to assess the knowledge about AMR among primary care pediatricians in the Lombardy region in Italy. The secondary aim is to evaluate the influence of non-clinical factors on antibiotic prescription behavior.

## 2. Materials and Methods

This was a cross-sectional study based on a 32-item questionnaire performed between 1 April and 30 September 2023.

The questionnaire was developed by a panel of two pediatric infectious disease specialists and two experts in public health. The questionnaire was then pilot-tested. First, it was sent to three pediatricians to highlight possible unclear questions or answers. After considering their comments, the questionnaire was sent to two other pediatricians, who were asked to fill it out twice within a 10-day interval. The intra-rater reproducibility was found to be 97%.

After the development of the questionnaire and its pilot test, the link to the final questionnaire (full version available in the online Appendix A) was provided to all participants of the Pediatric Conference of the Primary Care Pediatricians Society in Lombardy. Then, two reminders were sent by email to the members of the society after 3 and six months.

The first part of the questionnaire collected information on demographics, such as the participant’s sex and year of graduation, and their length of service. The second one was focused on participation in conferences on AMR and antimicrobial stewardship and 5 simulated clinical reports on common infections in pediatric patients, including laryngitis, bronchiolitis, bronchitis, group A streptococcal (GAS) pharyngotonsillitis and acute otitis media. These clinical reports described a medical condition suggesting several therapies based on antibiotic treatment or clinical observation and antalgic therapy, such as painkillers. Each clinical case asked for only one answer, resulting in the need to choose the most appropriate response. These clinical questions, combined with the questions on antibiotic stewardship, aimed to assess the knowledge and attention towards AMR.

The third part collected information on the perceived relevance of the AMR frequency in the pediatric population, as well as on the potential influence of non-clinical factors, such as the fear of legal repercussions or parents’ insistence on antibiotic treatment for their child, on prescription behaviors. Data on AMR perceptions were collected through 5- or 6-point Likert scales.

Categorical data were analyzed as absolute frequencies and percentages. Data were compared using the Fisher exact test or chi-squared test, as appropriate. For inferential statistics, some answers were dichotomized as follows: influence of parental insistence (≤25% vs. >25% of antibiotic prescriptions) and fear of legal repercussions on antibiotic prescription behavior (Likert scale ≤2 vs. >2), diagnostic uncertainty (≤25% vs. >25% of antibiotic prescriptions). Multiple logistic regressions were then employed to investigate the association between the fear of legal repercussions and antibiotic prescription behavior or the tendency to prescribe antibiotics due to parental insistence (dependent variables) and the participants’ sex, length of service, and participation in conferences on the subject in the last three years (independent variables). A *p* < 0.05 was assumed as significant. The statistical language R, Vienna, version 3.5.3 (11 March 2019), was used for analysis.

In accordance with the decree of the Italian Ministry of Health of 30 January 2023, which does not identify observational non-clinical studies as requiring an ethical committee’s opinion, no submission to an ethical committee was required. Moreover, the collection of anonymous data with questionnaires not sent by e-mail but collected by paper or through a link is compliant in Italy with the GDPR and does not require approval through an ethics committee. 

The online or paper questionnaire was only completed by the subject if he/she agreed to participate in the survey.

## 3. Results

Despite obtaining a total of about one thousand pediatricians, only 253 of them completed the questionnaire for a response rate close to 24%.

In Table 1, the demographic data of the pediatricians responding to the questionnaire are reported. Most respondents were female (82%) and graduated between 1981 and 1990 (56%), with more than 10 years of experience working as a primary care pediatrician. 

Most respondents (85%) were board-certified in pediatrics, without any additional specialization.

The majority of the respondents expressed an interest in attending seminars and/or conferences on AMR (97.2%). About a quarter of the respondents (28.9%) had already participated in more than one conference or seminar on this topic during their professional career, while more than a third (36.0%) had attended at least one of them. 

Table 2 provides data about the social factors influencing antibiotic prescription. A total of 28 physicians (11.1%) identified parental pressure as a social factor influencing antibiotic prescription, whereas almost one third of the respondents identified diagnostic uncertainty as an influencing factor (32.2%). The knowledge gap and fear of legal repercussions, on the other hand, were both indicated by about 25% of the respondents as influencing factors in their prescribing activities.

Figure 1 presents data about the perception of the need for a national plan on AMR. A total of 174 pediatricians (71.6%) identified a strong necessity for this, indicated by ratings of five and six points on the Likert scale. Only 11 responders (4.5%) considered it unnecessary to address this issue.

Figure 2 reports the perceptions among pediatricians about the frequency of AMR in pediatric settings. Approximately half of the participants (41.6%) considered AMR uncommon in pediatric settings.

Figure 3 shows the responses to the five clinical cases investigated. In the case of group A streptococcal (GAS) pharyngotonsillitis, 98.8% of the respondents indicated treatment with antibiotics, while, for bronchitis and laryngitis, antibiotic prescription was considered as the first intervention in 42.3% and 12.2% of cases, respectively. More than 60% indicated antibiotic therapy as the first treatment for acute otitis media, whereas 37.9% indicated the need to assess the patient’s characteristics before embarking on antibiotic therapy.

Data on the factors associated with a stronger fear of legal repercussions and the influence of parental insistence on antibiotic prescription are shown in Table 3 and Table 4, respectively. Diagnostic uncertainty was associated both with a stronger fear of legal repercussions and the influence of parental pressure. The length of service, participation in conferences on the topic in the last three years, and sex were not associated with a fear of legal repercussions or the influence of parental pressure.

## 4. Discussion

This study represents one of the first attempts to analyze the reasons behind inappropriate antibiotic prescription in a pediatric primary care setting. The results indicate that AMR in pediatrics is likely under-evaluated by many primary care pediatricians in Lombardy. Furthermore, diagnostic uncertainty emerged as one of the main determinants of antibiotic prescription.

The inability to use tests that quickly determine the viral origin of an infectious disease was found to be one of the primary reasons behind inappropriate prescribing. The findings from this study, along with the existing literature, highlight the relevance of decision security over other variables [16,17,18] and further highlight the importance of developing and validating new diagnostic tools for the management of children with an acute infection [19,20,21,22,23,24].

Many previous studies have reported the influence of fear and defensive medicine on inappropriate antibiotic prescription [25,26,27,28,29,30]. A few other observations have pointed out the relevance of parental pressure as a primary driver in antibiotic prescription decisions [30,31,32]. Our study reinterprets these aspects, considering social factors and knowledge as one of several aspects rather than the exclusive causal element. On the other hand, diagnostic uncertainty was found to be relevant to both antibiotic prescription and other factors potentially associated with inappropriate prescription. This finding suggests that new efforts should be made to address the issue of accurate diagnosis in the context of primary care assistance. This issue is particularly important for primary care pediatricians, who may face uncommon clinical presentations and have limited or delayed access to the advanced laboratory diagnostics available in hospitals.

Several previous studies have focused on the knowledge of and training on AMR as the main factor behind proper prescription, investigating this especially among medical and healthcare workers and students [33,34,35,36,37,38]. In this study, we also evaluated the knowledge and perceptions of AMR among pediatricians. Regarding the first point, the response rate was close to 24%. This low rate could suggest low interest in AMR among primary care pediatricians, especially considering their perceptions of its frequency in pediatric settings. However, the responses regarding the necessity of a national plan to respond to AMR underscore the perception of AMR as one of the main problems for the healthcare system. We speculate that new educational interventions and programs focused on primary care could be relevant to increase the awareness of AMR.

A critical element to counterbalance the increase in AMR among children could be vaccination against bacterial pathogens responsible for specific diseases. The use of antibiotics like erythromycin or beta-lactams could be avoided for respiratory pathologies with available vaccines. A history of vaccination for particular pathogens may assist in differentiating between bacterial and viral infections; while not being a diagnostic tool, it would provide relevant information for clinical reasoning. For instance, studies have reported that vaccinations, including the recent COVID-19 vaccine, have reduced respiratory diseases and the risk of bacterial co-infections, thereby reducing antibiotic abuse [39,40,41,42,43,44,45]. Ensuring proper vaccination coverage would also guarantee herd immunity, reducing disease occurrence even in non-vaccinable populations and consequently reducing inappropriate antibiotic use [46].

The results of this study should be interpreted considering some limitations. The data were collected within a specific geographical area, potentially influencing the findings regarding the importance of social and cultural factors in prescribing practices. Expanding the study sample would enhance the generalizability of the results and account for different cultural contexts. Additionally, the questionnaire was pilot-tested but not validated, and the response rate was low, albeit similar to previous studies in the same population [47,48].

## 5. Conclusions

AMR will represent one of the main issues for public health in the next few years, requiring an ever-greater awareness not only of the problem itself but also of the causal factors. In this regard, knowledge of this phenomenon among the general population as well as healthcare workers (HCW) is crucial, as highlighted by the great number of studies in the literature aimed at assessing this element [49,50,51,52,53,54,55]. This study represents a significant contribution to the existing literature, simultaneously examining a range of factors related to both the subject matter and the factors traditionally associated with over-prescription. 

This study highlights the relevance of diagnostic uncertainty in antibiotic prescription among primary care pediatricians. The findings might open up a new interpretation of the role of the defensive approach in prescribing practices and the development of AMR. This also necessitates a re-evaluation of the importance of a laboratory network that could enable the primary care pediatrician to promptly access tests that would enhance their prescribing autonomy. This study therefore represents a first step for future research, both nationally and in hospital settings, to confirm the findings and guide interventions to address this issue.

## Figures and Tables

**Figure 1 ijerph-21-01091-f001:**
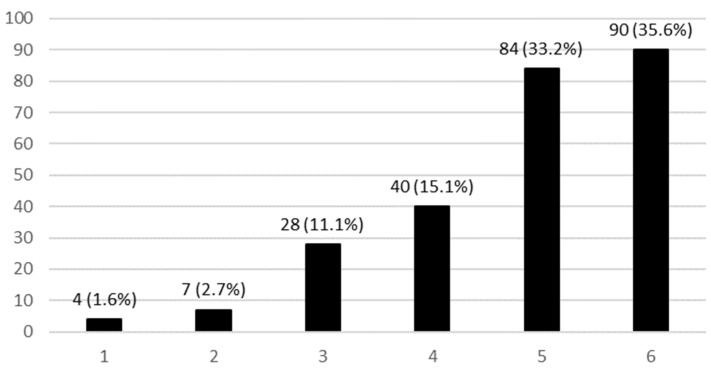
Perception of the need for a national AMR national plan in pediatric settings. The Likert scale indicates 1—not necessary, 2—not particularly necessary, 3—necessary, 4—very necessary, and 5—essential.

**Figure 2 ijerph-21-01091-f002:**
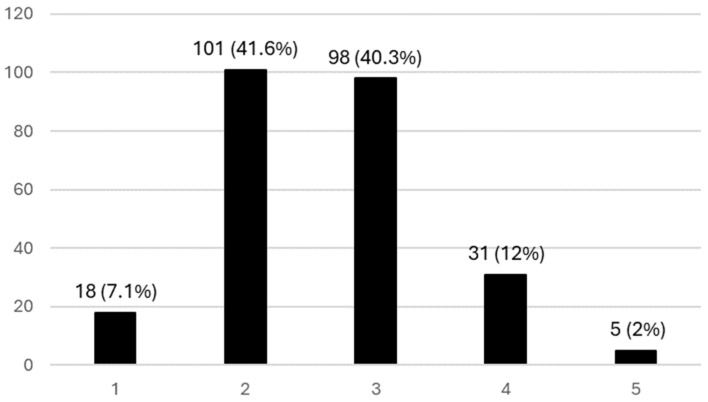
Perceptions about the frequency of AMR in pediatric settings. The Likert scale indicates 1—rare, 2—infrequent, 3—neither frequent nor infrequent, 4—very frequent, and 5—extremely frequent.

**Figure 3 ijerph-21-01091-f003:**
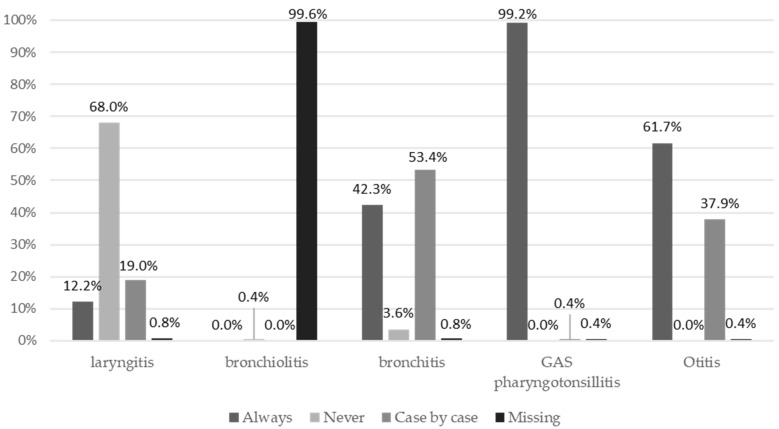
The frequency of pediatricians who would prescribe antibiotics for different pediatric infectious diseases.

**Table 1 ijerph-21-01091-t001:** Demographic data of respondents (N = 253).

Sex	
F	208 (82.2%)
M	45 (17.8%)
Year of graduation	
1960–1970	2 (0.8%)
1971–1980	17 (6.7%)
1981–1990	143 (56.5%)
1991–2000	70 (27.7%)
2001–2010	15 (5.9%)
2011–2020	6 (2.4%)
Year of post-graduate specialization	
1971–1980	3 (1.2%)
1981–1990	81(32.0%)
1991–2000	122 (48.2%)
2001–2010	31 (12.3%)
2011–2020	16 (6.3%)
Years spent working as a primary care pediatrician	
1–5 years	9 (3.6%)
5–10 years	12 (4.7%)
>10 years	232 (91.7%)
Further specialization	
Yes	38 (15.0%)
No	215 (85.0%)
Participation in seminars and/or conferences on antimicrobial resistance	
No, and no interest in following them	7 (2.8%)
No, but with interest in following them	82 (32.3%)
Yes, but only 1	91 (36.0%)
Yes, and more than 1	73 (28.9%)

**Table 2 ijerph-21-01091-t002:** Factors influencing the decision to prescribe antibiotics (N = 253).

	N (%)
Knowledge gaps	
0	194 (76.7%)
1	59 (23.3%)
Legal repercussions	
0	189 (74.7%)
1	64 (25.3%)
Parental pressure	
0	225 (88.9%)
1	28 (11.1%)
Diagnostic uncertainty	
0	170 (67.2%)
1	83 (32.8%)

**Table 3 ijerph-21-01091-t003:** Results from multiple logistic regression on predictors of greater fear of legal repercussions (dependent variable) in antibiotic prescription.

Variable	Odds Ratio	95% CI	*p*-Value
Courses in the last 3 years (no, but I would like to follow them)	0.243	0.04–1.31	0.1000
Courses in the last 3 years (yes)	0.210	0.04–1.09	0.0633
Length of service (5–10 years)	0.459	0.06–3.22	0.4330
Length of service (>10 years)	0.444	0.10–1.86	0.2670
Sex (male)	0.815	0.36–1.81	0.6150
Diagnostic uncertainty (1)	3.970	2.15–7.30	0.0001

**Table 4 ijerph-21-01091-t004:** Results from multiple logistic regression on predictors of stronger influence of parental pressure (dependent variable) in antibiotic prescription.

Variable	Odds Ratio	95% CI	*p*-Value
Courses in the last 3 years (no, but I would like to follow them)	1.030	0.10–9.79	0.9760
Courses in the last 3 years (yes)	0.596	0.06–5.50	0.6480
Years of working (5–10 years)	0.759	0.08–7.03	0.8080
Years of working (>10 years)	0.464	0.08–2.46	0.3670
Sex (male)	0.887	0.30–2.59	0.8270
Diagnostic uncertainty (1)	2.310	1.03–5.22	0.0433

## Data Availability

Dataset available on request from the authors.

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
