# Peer review of "Factors behind Antibiotic Therapy: A Survey of Primary Care Pediatricians in Lombardy"

_ijerph, 2024, doi:10.3390/ijerph21081091_

Round 1

Reviewer 1 Report

Comments and Suggestions for Authors

The authors have conducted an excellent study with the objective of investigating the factors underlying antibiotic prescriptions by pediatricians in the Lombardy region. The study achieved a response rate of approximately 24%, highlighting the necessity for interventions to prevent this phenomenon, as indicated by 71.6% of respondents. Nonetheless, about half of the respondents considered the phenomenon of antimicrobial resistance (AMR) to be uncommon in pediatric settings. Among the identified risk factors, diagnostic uncertainty was found to be significantly more influential than parental pressure (OR 2.31, CI 95% 1.03–5.22) and fear of legal repercussions (OR 3.97, CI 95% 2.15–7.30), with the inability to diagnose the bacterial origin of an infection emerging as the primary driver of prescribing choices over non-clinical factors such as parental demands or fear of lawsuits.

However, some minor revisions or justifications are required:

1. Why did the authors select only a six-month study duration?

2. How do the authors justify their conclusions based on a 24% response rate/sample size?

3. Authors are requested to include references to previous studies in their description section.

--------------

Author Response

The authors have conducted an excellent study with the objective of investigating the factors underlying antibiotic prescriptions by pediatricians in the Lombardy region. The study achieved a response rate of approximately 24%, highlighting the necessity for interventions to prevent this phenomenon, as indicated by 71.6% of respondents. Nonetheless, about half of the respondents considered the phenomenon of antimicrobial resistance (AMR) to be uncommon in pediatric settings. Among the identified risk factors, diagnostic uncertainty was found to be significantly more influential than parental pressure (OR 2.31, CI 95% 1.03–5.22) and fear of legal repercussions (OR 3.97, CI 95% 2.15–7.30), with the inability to diagnose the bacterial origin of an infection emerging as the primary driver of prescribing choices over non-clinical factors such as parental demands or fear of lawsuits.

However, some minor revisions or justifications are required:

Comment 1. Why did the authors select only a six-month study duration?

Response1. We thank the reviewer for this question. The six months was due to two reminders sent by email. We added this information in the revise version of the manuscript.

Comment 2. How do the authors justify their conclusions based on a 24% response rate/sample size?

Response 2. We really appracited the reviewer comment. We have tone downed our conclusion and add the low sample size in the revised manuscript. On the opther hand we highlighted that similar response rates were obtained in other surveys in similar populations ((Respir Med. 2024 May:226:107587. doi: 10.1016/j.rmed.2024.107587. Epub 2024 Mar 24., Sapienza M, Furia G, La Regina DP, Grimaldi V, Tarsitano MG, Patrizi C, Capelli G; Rome OMCeO Group; Damiani G. Primary care pediatricians and job satisfaction: a cross sectional study in the Lazio region. Ital J Pediatr. 2023 Aug 25;49(1):104. doi: 10.1186/s13052-023-01511-x)

Comment 3. Authors are requested to include references to previous studies in their description section.

Response 3 We thank the reviewer. In the revised version of the manuscript wr have included new references

Reviewer 2 Report

Comments and Suggestions for Authors

The title of the manuscript is very interesting to the readers. 

The Abstract is an accurate summary of the research.

The Introduction very consistent. 

Objective

The main objetive is poorly crafted. In my opinion, the article is not analyzing knowledge of AMR throughout the country, but  in the Lombardy Region. Therefore, I do  not know what the authors intend to achieve in this research. 

Methodology

Some serious flaws have been detected:

(i) The authors do not mention the validation of the questionnaire in the manuscript. Thus, I do not know if this  instrument  has been useful. 

(ii) If the questionarie has been validated, how has it tested?  it was not mentioned in the manuscript.

(iii) There is no homogeneity in the sample by  sex. It could  lead to a bias. 

(iv)  In addition, the value attributed to each replies from the questionnaire  has not been sufficiently explained. I honestly think that such information should be shown in methodology section. 

Thus, the reporting of the methods have not been sufficiently detailed so that the research might be reproduced. 

Thus, the reporting of the methods have not been sufficiently detailed so that the research might be reproduced

Results 

According to the main objective, I miss data from the rest of region of Italy.

 Even though, authors have described the years of experience in table 1, No relationship has been established between years of experience and its influence on prescription. Adressing this issue would have enhanced the value of the article.

Discussion 

Sentences too long. Please, revise the grammar estructures.

Lines 157-158. As regards this statement.  In my opinión, there are many studies that analyze non clinical factor related to inappropiate use of antibiotics.

Conclusions 

The conclusion of this study does not provide relevant information related to the main objective. 

Ethical Considerations

Since the sample of this  research is human beings, the study must have been approved by the Clinical Research Ethics committee.

Comments on the Quality of English Language

The article should be revised by the english editing service: long sentences and too many conectors . For instance:  Lines 215-219

Author Response

The Introduction very consistent. 

Objective

Comment 1. The main objetive is poorly crafted. In my opinion, the article is not analyzing knowledge of AMR throughout the country, but  in the Lombardy Region. Therefore, I do  not know what the authors intend to achieve in this research. 

Response 1. We thank the reviewer for this comment. We have amended the text to indicate the area of interest and research in the region of Lombardy, Italy.

Methodology

Some serious flaws have been detected:

Comment 2. The authors do not mention the validation of the questionnaire in the manuscript. Thus, I do not know if this  instrument  has been useful. 

Response  2. We thank the reviewer again for this important point. The questionnaire was developed by a panel of experts and then pilot tested. This process is now dewscribed in the revised version of the manuscript. The lack of a formal validation is now listed among limitations in the revise version of the manuscript.

Comment 3. If the questionarie has been validated, how has it tested?  it was not mentioned in the manuscript.

Response  3. Thank you. As previously addressed the questionnaire was pilot tested and interrater variability was assessed.

Comment 4. There is no homogeneity in the sample by  sex. It could  lead to a bias. 

Response  4. We appraciateb this comment. Actually most of primary care pediatricians in Italian are female. Other studies on the same population provided similar sample chaacteriostics (Respir Med. 2024 May:226:107587. doi: 10.1016/j.rmed.2024.107587. Epub 2024 Mar 24., Sapienza M, Furia G, La Regina DP, Grimaldi V, Tarsitano MG, Patrizi C, Capelli G; Rome OMCeO Group; Damiani G. Primary care pediatricians and job satisfaction: a cross sectional study in the Lazio region. Ital J Pediatr. 2023 Aug 25;49(1):104. doi: 10.1186/s13052-023-01511-x)

Comment 5.  In addition, the value attributed to each replies from the questionnaire  has not been sufficiently explained. I honestly think that such information should be shown in methodology section. 

Response  5. No scores were assigned to the various questions with the intention of describing the awarenessnees and knowledge of the phenomenon rather than quantifying the level of preparedness of the professionals investigated.

Comment 6. the reporting of the methods have not been sufficiently detailed so that the research might be reproduced. 

Response  6. We have tried to further detail the methodology of the study. We have also included a copy of the questionnaire used in the study as supplementary materials.

Results 

Comment 7. According to the main objective, I miss data from the rest of region of Italy.

Response  7. We have changed the description of the aim, as it was misleading, by specifying the survey on the population of pediatricians in the Italian region of Lombardy.

Comment 8. Even though, authors have described the years of experience in table 1, No relationship has been established between years of experience and its influence on prescription. Adressing this issue would have enhanced the value of the article.

Response 8. Thank you for this request. The influence of years of experience has already been analyzed in the context of logistic regressions showing that it has no significant influence in this area.

Discussion 

Comment 9. Sentences too long. Please, revise the grammar estructures.

Response 9. We agree with the reviewer. We have revised the sentence accordingly

Comment 10. Lines 157-158. As regards this statement.  In my opinión, there are many studies that analyze non clinical factor related to inappropiate use of antibiotics.

Response 10. We have already changed the text due you suggestion.

Conclusions 

Comment 11. The conclusion of this study does not provide relevant information related to the main objective. 

Response 11.We have already rewritten conclusion to ensure that information is provided both in greater detail and in a clearer and more direct manner.

Ethical Considerations

Comment 12. Since the sample of this  research is human beings, the study must have been approved by the Clinical Research Ethics committee.

Response 12. The collection of anonymous data with questionnaires by paper or through a link is compliant in Italy with the articles of the gdpr and does not require passage through an ethics committee in Italy.

Reviewer 3 Report

Comments and Suggestions for Authors

This is an interesting article dealing with reasons behind eventual inappropriate antibiotic prescribing among paediatricians in Lomardy, Italy.

However there are few suggestions for the author to improve the manuscript. 

In the summary, tlines 15-17 need english improvement. Also, the results are nor well presented. Please focus on the most important results of the study and present them clear and concise. This sentence in particular need improvement: "The study saw a response rate of approx- 20 imately 24% and a high focus on the need for interventions to prevent the phenomenon (71.6%)."

In the introduction, lines 45-47 need reference to confirm the statement for antibiotic prescribing in Italy. Also, it is unclear do the authors investigate paediatritian in all Italy or just region Lombardy. Please specify in Aims. 

In the Material and methods there is no explanation how is AMR knowladge assesed?

Lines 80-82 are repetition. 

sentence in line 82: "Categorical data as absolute frequency and percentage". has no meaning, please correct.

Discussion and conclusion need extensive changes. 

Discussion needs to start with the most interesting finding In this case that would be major reason for antibiotic prescribing: uncertanty in diagnosis. Then discussing it and comparing with different studies of the same topic. 

Lines 144-148 should be the closure of the conclusion, not the begining. 

In contrary, lines 221-231, which are begining of the Conclusion section should be the first paragraph of Discussion. 

Limitations should been Discussion not Conclusion. 

And Conclusion should answer to the proposed question of the authors in the Introduction. 

Legends of the Figures should be improved, self explaining and also marking what is representing.  

Comments on the Quality of English Language

English has some minor faults. Please revise. 

Author Response

This is an interesting article dealing with reasons behind eventual inappropriate antibiotic prescribing among paediatricians in Lomardy, Italy.

However there are few suggestions for the author to improve the manuscript. 

Comment 1. In the summary, tlines 15-17 need english improvement. Also, the results are nor well presented. Please focus on the most important results of the study and present them clear and concise. This sentence in particular need improvement: "The study saw a response rate of approx- 20 imately 24% and a high focus on the need for interventions to prevent the phenomenon (71.6%)."

Response 1. We thank the reviewer for pointing this out. We have rewritten the summary, especially the results section, and extensively revised the results section.

Comment 2. In the introduction, lines 45-47 need reference to confirm the statement for antibiotic prescribing in Italy. Also, it is unclear do the authors investigate paediatritian in all Italy or just region Lombardy. Please specify in Aims. 

Response  2. We have put the reference and clarified the population in the aim, indicating Lombardy pediatricians as the population.

Comment 3. In the Material and methods there is no explanation how is AMR knowladge assesed?

Response  3. We have added an explanation of the way in which knowledge of AMR was assessed, bearing in mind that the project intended awarenessnees and knowledge of the phenomenon as knowledge rather than quantifying the level of preparedness of the professionals investigated.

Comment 4. Lines 80-82 are repetition. 

Response 4. Thank you. We deleted the repetition.

Comment 5. sentence in line 82: "Categorical data as absolute frequency and percentage". has no meaning, please correct.

Response 5. We thank for this comment and revised the manuscript accordingly.

Discussion and conclusion need extensive changes. 

Comment 6. Discussion needs to start with the most interesting finding In this case that would be major reason for antibiotic prescribing: uncertanty in diagnosis. Then discussing it and comparing with different studies of the same topic. 

Response 6. We rewrote the first part of the discussion according to the reviewer’s suggestions and focused on diagnostic uncertainty. At the same time, we also discussed the results of the present study in view of previous articles.

Comment 7. Lines 144-148 should be the closure of the conclusion, not the begining. 

Response 7. We modified the manuscript according to this suggestion

Comment 8. In contrary, lines 221-231, which are begining of the Conclusion section should be the first paragraph of Discussion. 

Response 8. We modified the manuscript according to this suggestion.

Comment 9. Limitations should been Discussion not Conclusion. 

Response 9. We agree with the reviewer and modified the sections accordinlgy

Comment 10. And Conclusion should answer to the proposed question of the authors in the Introduction. 

Response 10. We have changed conclusion to answer to main question reported in the Introduction.

Comment 11. Legends of the Figures should be improved, self explaining and also marking what is representing.  

Response 11. We have expanded the figure legends as suggested.

Round 2

Reviewer 2 Report

Comments and Suggestions for Authors

After making relevant modifications in the manuscript, the scientific soundness has been enhanced. 

However, I would like to highlight with regard to ethical issues , I honestly think that this study  would have required the  Institutional ethical Review Board  approval 

Author Response

Comments 1: After making relevant modifications in the manuscript, the scientific soundness has been enhanced.

Response 1: I am grateful for your kind reply and your appreciation of the changes.

Comments 2: However, I would like to highlight with regard to ethical issues , I honestly think that this study  would have required the  Institutional ethical Review Board  approval

Response 2: In response to your request, we conducted a comprehensive review of the relevant legislation. Our analysis indicated that, in accordance with the ministerial decree of 30th January 2023, this particular type of study, which is a non-clinical observation, does not fall within the scope of studies that necessitate an opinion from the ethics committee. In addition,  the anonymous collection of data is in compliance with the GDPR regulation and does not necessitate the involvement of the ethics committee with regard to the privacy of respondents.

Reviewer 3 Report

Comments and Suggestions for Authors

Authors have improved manuscript according to all suggestions. 

Author Response

Comments 1 Authors have improved manuscript according to all suggestions. 

Response 1 I am truly grateful for your thoughtful response. 
